# Assessing Reduction of Antibiotic Prescribing for Acute, Non-Complicated Infections in Primary Care in Germany: Multi-Step Outcome Evaluation in the Cluster-Randomized Trial ARena

**DOI:** 10.3390/antibiotics10101151

**Published:** 2021-09-24

**Authors:** Regina Poss-Doering, Dorothea Kronsteiner, Martina Kamradt, Petra Kaufmann-Kolle, Edith Andres, Veit Wambach, Julian Bleek, Michel Wensing, Joachim Szecsenyi

**Affiliations:** 1Department of General Practice and Health Services Research, University Hospital Heidelberg, Im Neuenheimer Feld 130.3, 69120 Heidelberg, Germany; martina.kamradt@med.uni-heidelberg.de (M.K.); michel.wensing@med.uni-heidelberg.de (M.W.); joachim.szecsenyi@med.uni-heidelberg.de (J.S.); 2Institute of Medical Biometry, University Hospital Heidelberg, Im Neuenheimer Feld 130.3, 69120 Heidelberg, Germany; kronsteiner@imbi.uni-heidelberg.de; 3aQua Institute, Maschmuehlenweg 8-10, 37073 Goettingen, Germany; petra.kaufmann-kolle@aqua-institut.de (P.K.-K.); edith.andres@aqua-institut.de (E.A.); 4Agentur Deutscher Arztnetze e.V., Friedrichstraße 171, 10117 Berlin, Germany; info@drwambach.de; 5AOK-Bundesverband, Rosenthaler Straße 31, 10178 Berlin, Germany; julian.bleek@bv.aok.de

**Keywords:** antimicrobial resistance, antibiotic prescribing, acute non-complicated infections, primary care, data-based feedback, mixed logistic regression model, multi-faceted intervention

## Abstract

The three-armed cluster-randomized trial ARena (sustainable reduction of antibiotic-induced antimicrobial resistance) aimed to foster appropriate antibiotic use and reduce overprescribing in German ambulatory care to counter antibiotic resistance. Multi-faceted interventions targeted primary care physicians, teams and patients. This study examined the effectiveness of the implementation program. ARena was conducted in 14 primary care networks with 196 practices. All arms received data-based feedback on antibiotics prescribing and quality circles. Arms II and III received different add-on components each. Primary outcome examined is the prescribing rate for systemic antibiotics for cases with non-complicated acute infections (upper respiratory tract, bronchitis, sinusitis, tonsillitis, otitis media). Secondary outcomes refer to the prescribing of quinolones and guideline-recommended antibiotics. Based on pseudonymized quarterly claims data, mixed logistic regression models examined pre-post intervention antibiotic prescribing rate changes and compared to matched standard care. A significant rate reduction (arm I 11.7%; arm II 9.9%; arm III 12.7%) and significantly lower prescribing rates were observed for all arms (20.1%, 18.9% and 23.6%) compared to matched standard care (29.4%). Fluoroquinolone prescribing was reduced in all intervention arms and rates for recommended substances generally increased. No significant post-interventional difference between intervention arms was detected. Findings indicate implementation program impact compared to standard care.

## 1. Introduction

On a global and on national scale, the development and increase of antimicrobial resistance are advanced by over-consumption of antibiotics in human and veterinary healthcare and by inappropriate prescribing. In Germany, about 85% of the used antibiotics are prescribed during consultations in the primary care sector [1,2]. Compared to antibiotic consumption in other countries, the use of antibiotics in Germany is considered moderate [3,4]. However, there is a potential for a lowering of prescribing rates for specific conditions treated in primary care, where antibiotics are prescribed by general practitioners in 41% of all consultations for acute respiratory tract infections (ARTI) [2] and only 52% of these prescriptions correspond to prevailing clinical recommendations [5].

In the context of the national strategy to reduce antibiotic consumption and counteract antimicrobial resistance [1], the ARena trial (sustainable reduction of antibiotic-induced antimicrobial resistance) was conducted as a (non-blinded) three-armed cluster randomized trial between 2017 and 2020. The aim of the trial was to examine the impact of a complex multifaceted implementation program intended to foster the rational use of antibiotics and reduce antibiotic prescribing for acute non-complicated infections of viral origin in primary care settings. The comprehensive program was developed in collaboration with the participating primary care networks (PCNs) and based on relevant published research identified in a literature review [6,7,8,9,10]. PCNs can be understood as formalized collaborations of healthcare providers who interact on a regular basis, standardize care, share patients, discuss concerns, support practice management and attend continuing education [11]. Such collaborative healthcare provider networks support coordination of care and contribute to improvements in care quality and safety [12]. Thus, they were expected to amplify the impact of the implementation program.

The ARena implementation program used multi-faceted interacting components to address knowledge, attitudes and experiences of primary care physicians (General Practitioners, Gynecologists, Otolaryngologists, Urologists, and Pediatricians), care teams and patients about the use of antibiotics in PCNs across two German federal states (Bavaria and North Rhine-Westphalia). The program used different strategies to reduce undesirable prescribing rates. All three intervention arms received a base module of intervention components comprising e-learning on communication for physicians to limit misinterpretation of patient preferences, moderated quality circles (QCs) with data-based feedback on antibiotics prescribing for physicians to facilitate appropriate prescribing, public information campaigns including social media, performance-based additional reimbursement, and culture-sensitive print information material for patients. Based on program developers’ assessment of barriers for implementation, arm II received e-learning on communication and separate QCs addressing the team of medical assistants in participating practices, and tablet pcs providing patient information in waiting areas. Including the medical assistants was assumed to be supportive for rational prescribing since sensitizing practice teams for a stronger interaction with patients might facilitate a relieve of the physician’s workload and contribute to optimized decision-making [13,14]. Considering the developments in German healthcare, add-ons in arm III were future-orientated and comprised of multidisciplinary, cross-sectoral QCs and a computerized decision support system (CDSS) to be used in the practice admin system. All QCs were offered to promote assessment and critical discussion of clinical practice and focused on care quality and appropriate use of antibiotics regarding respiratory tract infections, urinary tract infections, community acquired pneumonia and multi-resistant pathogens. Current evidence of all base module intervention components validated positive effects on physician prescribing [8,15,16,17]. Evidence regarding effects of add-on components in arms II and III was still limited [16,18,19], or inconclusive [16,20,21]. Nevertheless, effects were expected to be higher in arms II and III than in arm I, since add-ons targeted specific barriers for change and were assumed to be promising tools for reducing errors in medical decision-making [22]. A more detailed description of the study design and intervention components used can be found elsewhere [18].

The primary objective of this study was to examine and compare the change of prescribing rates for targeted index diagnoses in the ARena intervention arms and compare prescribing rates to non-participating standard care, matched to participating PCNs, based on quarterly claims data.

## 2. Results

A total of 87,377 patient cases with ARena index diagnoses during the intervention period (Q4 2017-Q2 2019) were included for analysis, which exceeded the target of 79,685 cases calculated for the evaluation. Sample sizes differ between primary and secondary outcomes, because they focus on subgroups of the primary outcome population. One patient might represent several cases.

Physicians working in general practice were the largest group of prescribers in PCNs (76.5%) and in standard care (75.8%) (Appendix A details medical specialty groups distribution). Study populations differed per indexed consultation reason regarding number of cases and age of patients. The sample sizes in the three intervention arms were as follows: in intervention arm I pre 9673 and post 10,143 cases, in intervention arm II pre 4583 and 6730 post cases, and in intervention arm III pre 3951 and post 5076 cases. For the matched standard care cohort, 25,385 cases pre and 25,144 post-intervention were included. Patient characteristics for the included cases are presented in Table 1 (see Appendix A for a detailed description pre- and post-intervention).

### 2.1. Primary Outcome

Intervention arms showed different baseline prescribing rates and post-intervention prescribing rate reductions. A significant reduction of antibiotic prescriptions was observed in intervention arm II (post versus pre: OR = 0.547 95%-CI = [0.493;0.607], *p* < 0.001) and in intervention arm III (post versus pre: OR = 0.519 95%-CI = [0.467;0.576], *p* < 0.001). Additionally, in intervention arm I, the pre-post-comparison showed a significant reduction of antibiotic prescriptions (OR = 0.523 95%-CI = [0.485;0.563], *p* < 0.001). No significant difference was detected between intervention arms regarding within-group change in the RCT. Figure 1 illustrates the primary outcome as percentages of cases with index diseases in the intervention arms and matched standard care (pre vs. post).

The comparison of changes in each of the three intervention arms to the matched standard care was evaluated in logistic mixed regression models and considered pre/post intervention as covariate. For all intervention arms, a larger reduction of antibiotic prescribing compared to the matched standard care was observed (11.7% in arm I vs. 4.5% in standard care: OR = 0.596 95%-CI = [0.572;0.621], *p* < 0.001; 9,9% in arm II vs. 4.5% in standard care: OR = 0.661 95%-CI = [0.629;0.695], *p* < 0.001; 12.7% in III vs. 4.5% in standard care: OR = 0.726 95%-CI = [0.689;0.764], *p* < 0.001)). Table 2 describes the pre- post comparison of antibiotic prescribing rates, as well as the pre-post comparison between intervention arms and Table 3 the comparison to the matched standard care adjusted for additional covariates.

Adjusting the primary outcome model for further patient and practice characteristics indicated that the difference between interventions arms and matched standard care remained significant. Female gender, age, and Charlson index were associated with a higher likelihood for an antibiotic prescription. Specialists prescribed less antibiotics compared to general practitioners. Table 3 details the comparison of intervention arms to the matched standard care adjusted for additional covariates.

### 2.2. Secondary Outcomes

For the secondary outcome (SO) of fluoroquinolone prescribing, low pre-interventional rates were observed for trial participants, and these were further reduced during the intervention period (for details see Appendix A). At the same time, the prescribing rate of fluoroquinolones decreased in standard care as well. The comparison between intervention arms and matched standard care showed that fluoroquinolone prescribing was lower in all intervention arms (I vs. standard care: OR = 0.911 95%-CI = [0.811;1.021]; II vs. standard care: OR = 0.71 95%-CI = [0.594;0.844]; III vs. standard care: OR = 0.716 95%-CI = [0.613;0.833]).

Prescribing of guideline-recommended substances was evaluated for each index disease separately as percentage of patient cases who consulted primary care practices with one of the index diseases and required and received antibiotics. The highest prescribing rate of recommended substances for URTI was observed at baseline (pre-intervention) in matched standard care. At the post-intervention measuring point, prescribing of guideline-recommended antibiotics in the intervention arms had increased and was higher than in matched standard care (I vs. standard care: OR = 1.615 95%-CI = [1.435;1.816]; II vs. standard care: OR = 1.456 95%-CI = [1.254;1.687]; III vs. standard care: OR = 1.539 95%-CI = [1.336;1.769]).

For bronchitis, an increase in prescribing of recommended substances was recognized in all intervention arms and in matched standard care. Intervention arms I and III reached a substantially higher rate compared to standard care (I vs. standard care: OR = 1.634 95%-CI = [1.439;1.855]; II vs. standard care: OR = 0.987 95%-CI = [0.83;1,17]; III vs. standard care: OR = 1.374 95%-CI = [1.181;1.594]). For sinusitis, all intervention arms showed higher post-interventional prescribing of recommended antibiotics (I vs. standard care: OR = 1.338 95%-CI = [1.057;1.691]; II vs. standard care: OR = 1.372 95%-CI = [1.012;1.851]; III vs. standard care: OR = 2.198 95%-CI = [1.685;2.859]). Patients suffering from tonsillitis in intervention arm I had the highest chance of receiving a recommended substance for the pre timepoint (33.5% recommended antibiotics), while the highest absolute increase was observed in intervention arm III (pre 10.7%, post 25.7%). Comparing post-interventional prescribing, only intervention arm I shows a higher prescribing rate of recommended antibiotics compared to matched standard care (I vs. standard care: OR = 1.157 95%-CI = [0.95;1.406]; II vs. standard care: OR = 0.682 95%-CI = [0.535;0.864]; III vs. standard care: OR = 0.685 [0.503; 0.919]). The highest prescribing rates of antibiotics recommended for otitis media are 41.3% of observed cases in intervention arm II post the intervention period (pre: 33.3%) and 37.9% in intervention arm I (pre: 27.7%), compared to 29.3% in matched standard care (pre: 23.6%).

Figure 2 illustrates the percentage of patient cases who consulted participating PCN practices for one of the index diseases and received a prescription for fluoroquinolones or were treated with indication-specific guideline-recommended systemic antibiotics. A decrease was targeted for fluoroquinolone prescribing while for recommended antibiotics, an increase in prescribing was targeted for cases that required antibiotics. Appendix A describes the observed changes in recommended antibiotics prescribing regarding the examined index diagnoses and indicates the size of the respective patient populations. Comparison of the intervention arms and to matched standard care regarding the SOs is detailed in Appendix A.

## 3. Discussion

The implementation program focused on the reduction of antibiotic prescribing in primary care in Germany for non-complicated self-limiting infections without indication for antibiotics. A significant reduction was observed in the pre-post comparison of all intervention arms. No significant difference between the randomized intervention arms was found, so add-on intervention components in arms II and III did not increase the effect. However, the implementation program was associated with lower antibiotic prescribing than in matched standard care, suggesting that it had impact on professional practice.

Different strategies were used to aim for a reduction of undesirable prescribing including enabling measures such as feedback reports, thematically focused quality circles and e-learning modules on physician-patient communication. Enabling measures have been shown to be associated with better acceptance and combining them with restricting interventions was associated with sustainability of the latter [19]. For the ARena trial, a combination of enabling and potentially restrictive intervention components was applied. Effects were expected to be higher in intervention arms II and III than in arm I since the add-on components targeted specific barriers for change (arm II) and match with overall trends in healthcare (arm III). Enabling intervention components in arm II were not only targeted at prescribing physicians directly, but also addressed the practice teams, assuming their role in ambulatory care could be supportive for reduced prescribing by transferring knowledge and awareness to patients. Findings of the process evaluation conducted alongside ARena indicate that physicians in arm II viewed the additional components, particularly the involvement of the practice teams, as very valuable, since they perceived an increased health literacy among the more sensitized patients and a decrease in patient demand for antibiotics [20,21,23]. Providing health-related knowledge to patients via a tablet pc app, however, was perceived to potentially discriminate against older patients with a lesser digital affinity, pose hygiene-related challenges in the practice and increase the practice team’s workload. These factors, and a general concern about potential theft of the provided tablet pcs, accounted for a somewhat reduced intervention fidelity regarding this component in arm II. [20]. Combined with an acknowledged lower than initially expected patient demand for antibiotics, this might explain why the estimated potential impact on prescribing could not be observed.

In intervention arm III, the add-on components related to interprofessional care and the increased relevance of information technology solutions in healthcare. PCNs participating in arm III offered cross-sectoral QCs on own initiative and due to internal factors, not all initially planned meetings could take place. The CDSS offered in arm III was implemented into administrative practice software to provide adequate prompts when index diagnoses were coded, which could be ignored or turned off. Its implementation was delayed due to technical difficulties, resulting in a shortened usage period for less practices than expected [20]. The process evaluation found that practices participating in arm III received new impulses from the interdisciplinary QCs and considered the CDSS a support for the integration of knowledge into daily care routines and the choosing of indication-appropriate antibiotics. However, a lower intervention fidelity regarding these two add-on components caused by delayed or incomplete implementation may be typical for newer approaches and might illustrate why the expected additional effect could not be detected.

The trial was conducted in the particular setting of PCNs. Prior research had concluded that such collaborative health professional networks can contribute to improving healthcare quality and safety [12] and interventions delivered by peer community can be expected to be more successful than those delivered by agencies less connected to program recipients [24]. Thus, the PCNs were expected to be an enabling factor for change of prescribing routines. This was also supported by process evaluation findings [20,25] which confirmed that it was an appropriate choice to conduct the ARena trial in the setting of PCNs. Observed antibiotic prescribing in the intervention arms at baseline [26] was both, lower than expected and lower than in matched standard care. It also differed between the three arms yet could be reduced significantly. As prescribing rate reduction effects tend to be stronger when initial rates are high, this can be seen as confirmation for appropriateness of the study setting and the chosen improvement program.

For the SO of fluoroquinolone prescribing, moderate prescribing still declined in all interventions arms as well as in the matched standard care. However, the chance of inappropriate antibiotic prescribing was substantially lower for cases in all intervention arms compared to cases in matched standard care. Prescribing of indication-specific guideline-recommended antibiotics for the observed index diseases (URTI, bronchitis, sinusitis, tonsillitis, otitis media) improved in the intervention arms and was higher than in matched standard care, with the exception of prescribing for otitis media in intervention arm III. Otitis media is a very common disease in early childhood [27,28,29,30] and one of the most common indications for prescribing antibiotics for children [31,32], though evidence and national guidelines might also recommend observation and close follow-up [33,34,35,36,37]. It can be assumed that it might also be recommended that antibiotic prescriptions for otitis media be written mainly by paediatricians who were underrepresented in the ARena intervention arms, and hence the matched standard care. This potentially explains prescribing of recommended antibiotics remaining somewhat low, since beside a nationwide decline in antibiotic prescribing to primary care patients in all age groups between 2010 and 2018, strong improvements and sustainable change in paediatric prescribing patterns have been identified [38].

For patients with co-morbidities, current guidelines in Germany recommend considering antibiotic treatment for non-complicated infections [39,40]. Prescribing rates of recommended antibiotics increased in all intervention arms, but compared to matched standard care, the likelihood for patients with acute tonsillitis to receive a guideline-recommended antibiotic was lower in intervention arms II and III. Prior research indicated that a physician’s personal attitude might be a determining factor in whether a patient receives an antibiotic prescription, or not [41]. A recent study evaluated characteristics of high and low prescribers of antibiotics in German primary care by examining routinely collected claims data. The study confirmed previous findings of considerable differences in prescribing rates between physicians and concluded that due to the limitations of secondary data, further research including the linkage of primary and secondary data should aim to determine causal relationships [42]. However, the discrepancy in our findings cannot be explained by the analysis, yet among possible reasons could also be diagnostic uncertainties referring to younger patients, or recurrent incidence of the disease, known allergies in patients, or regional variations.

### Strengths and Limitations

The particular strengths of this study are the randomized evaluation design, high validity of claims data, low drop out of participating physicians, thorough consideration of excluded ICD-10-coded diagnoses and the additional detailed analysis referring to the type of antibiotics (recommended, quinolones). All cases meeting the inclusion/exclusion criteria were considered. The main limitation of the randomized trial was the randomization at a high aggregation level (14 PCNs) with a risk of by-chance differences. In the observational comparison, the intervention arms were compared to matched standard care, not to a control group of practices from the participating PCNs, since the network structure and peer exchange likely would have led to contamination. At pre-intervention, observed prescribing rates differed between the arms and were lower than expected. Limitations also include the absence of blinding and health-related outcome measures, and the restriction on ICD-10 codes and claims-related health insurance data where disease, patient, and practice information were limited. Direct connection between ICD-10 codes and date of antibiotic prescription was not possible via the provided claims data. Therefore, ICD-10 codes and antibiotic prescriptions were matched at quarter level, which introduces a potential bias.

## 4. Materials and Methods

### 4.1. Study Design

ARena was conducted as a prospective (non-blinded) three-armed cluster randomized trial in two German federal states (Bavaria and North Rhine-Westphalia), with fourteen PCNs as units of randomization. The study was conducted with an intervention period of 21 months, and a two-part evaluation: (a) an outcome evaluation based on quarterly claims-data provided by the statutory health insurance provider AOK and (b) a process evaluation based on surveys [18]. Examined were the comparison of baseline and post-interventional prescribing rates, and differences between the intervention arms and the non-participant matched standard care generated from claims data.

### 4.2. Study Population

Fourteen PCNs in Bavaria (12 PCNs) and North Rhine-Westphalia (2 PCNs) were recruited by the aQua Institut, Göttingen, Germany, for participation in the ARena trial, and randomized into arm I (4 PCNs), arm II (5 PCNs), and arm III (5 PCNs) by a statistician at the Institute for Medical Biometry, University Hospital Heidelberg, Germany. Randomization considered the number of practices in each PCN, and based on the expected number of practices, aimed for comparable numbers of practices in each intervention arm. Due to intervention components to be offered on PCN level, randomization on practice level was not feasible. For administrative reasons, the focus was on cases in patients insured by AOK health insurance and registered within a specific healthcare delivery program (defined by German law § 140a SGB V a.F. and § 140a Abs. 1. S. 2 Alt. 1 SGB V n.F). At baseline, approximately 40,000 patients with AOK health insurance were registered in 196 participating primary care practices in the 14 PCNs. Medical specialties covered by participating physicians included general practice (largest group), otolaryngology, paediatrics, urology and gynaecology. Observed cases comprised patients who consulted a physician in participating PCNs or in the matched standard care for one of the following reasons: acute upper respiratory tract infection (URTI), acute bronchitis, acute sinusitis, tonsillitis, otitis media, urinary tract infection (UTI), community acquired pneumonia. All diagnoses were based on physician-recorded ICD-10 codes and prescribing information in the administrative data provided by the health insurer for quarterly reimbursement periods, which were linked by the pseudonymized individual insurance number. Cross-validation of recorded codes with patient charts was not possible due to data confidentiality regulations. Each physician-recorded ICD-10 code for defined index diagnoses in the observed quarters represents a case, and each patient could produce multiple cases. Diagnoses for other diseases and diagnoses that warrant an antibiotic therapy were not considered. (Appendix A details diagnoses and related ICD-10 codes and Appendix A provides a list of disregarded diagnoses.) Note that prescribing information was derived from quarterly claims data provided by the health insurer. Patients were not actively recruited, but cases were automatically included when a participating physician claimed reimbursement for them from the health insurer. All participating physicians consented in the use of their claims data and signed a data release form. Due to data protection regulations, patients in North Rhine-Westphalia had to give additional written consent to their data being included for analysis.

### 4.3. Measures

The primary outcome references established indicators of the European Surveillance of Antimicrobial Consumption Network (ESAC-Net) [43], which were tailored to the specifics of ARena. It was defined as the percentage of patient cases with acute non-complicated infections receiving an antibiotic prescription without pathogen detection when consulting primary care practices (cases with acute bronchitis (age 18–75), sinusitis (>18 years), otitis media (>2 years), URTI (>1 year), or tonsillitis (>1 year)). Evaluation was based on quarterly claims data. The intervention period was planned for 24 months and reduced to 21 months (Q4 2017 to Q2 2019) due to delays in component development. As indicated by the study protocol [18], points of measurement were at baseline (Q3 2016–Q2 2017) and at the end of the intervention period (Q3 2018–Q2 2019). Diagnoses for streptococcal tonsilitis and other pathogen-caused acute forms of tonsillitis that warrant antibiotic therapy were not considered.

SOs examined in this study refer to the percentage of all observed cases with (SO 1) acute non-complicated infections receiving a quinolone prescription when consulting primary care practices (cases with acute bronchitis (age 18–75), sinusitis (>18), otitis media (>2), URTI (>1 year), or tonsillitis (age >1)); (SO 2) acute URTI (>1 year) receiving a prescription for recommended antibiotics (amoxicillin); (SO 3) acute bronchitis (age 18–75) receiving a prescription for recommended antibiotics (amoxicillin, doxycycline, macrolides); (SO 4) sinusitis (>18 years) receiving a prescription for recommended antibiotics (amoxicillin, cephalosporins 2nd Gen, doxycycline); (SO 5) tonsillitis (>1 year) receiving a prescription for recommended antibiotics (penicillin, cephalosporins, erythromycin); and (SO 6) otitis media (>2 years) receiving a prescription for recommended antibiotics (amoxicillin, cephalosporins 2nd Gen, erythromycin).

The ARena study protocol also defined SOs referring to cystitis and community-acquired pneumonia which were not considered here due to a very small numbers of cases. Specifications for patient age in all observed sub-groups adhere to study protocol [18]. All outcomes reported here relate to ICD-10 coded consultations in primary care and prescribing of systemic antibiotics, and indication-specific prescribing of currently guideline-recommended antimicrobials [39,40]. Recommended antibiotics were categorized based on existing evidence-based clinical guidelines developed by the German College of General Practitioners and Family Physicians (DEGAM) [40] and the Association of the Scientific Medical Societies in Germany (AWMF) [39]. (See Appendix A for currently recommended and alternative antibiotics.)

The following patient-related sociodemographic, disease, and treatment characteristics were provided in the claims data and, as indicated by the study protocol [18], included for analysis: age, sex, Charlson comorbidity index (CCI) (predicts 1-year survival in patients based on sum of relevant comorbidities) [44,45], nationality (missing values were aggregated to “other”). For the primary care practices, type of location (urban, urbanized, countryside), type of practice (single or group) and medical specialty group are documented.

### 4.4. Data Analysis

Data were analysed using the statistical software R version 3.6.3 to compare four baseline quarters (Q3 2016–Q2 2017) to four quarters at the end of the intervention (Q3 2018–Q2 2019) for the intervention arms, between arms, and in comparison to matched standard care regarding the defined outcomes. The primary and secondary outcomes, documented data referring to patient and disease characteristics, treatment data, and distribution of medical specialty group were first analysed descriptively stratified by the intervention arm. For categorical variables, absolute and relative frequencies are provided. Note that patient and disease characteristics, treatment data and practice characteristics differ between outcomes, as considered cases are defined for each outcome by the respective disease and prescribing of antibiotics. Confirmatory analysis of the primary outcome was conducted based on the Intention-To-Treat (ITT) population. Since the pre/post reduction was of primary interest and reduction was assumed to be different in each intervention arm, a structured three-step, hierarchical test procedure was applied [46] to ensure a global significance level of 5% and assess the following hypotheses regarding the pre-post intervention comparison of the percentage of cases with acute non-complicated infections receiving antibiotic prescription when consulting primary care practices:Step 1: Percentage of cases in intervention arms II and III are each lower post-intervention.Step 2: Percentage of cases in intervention arm I is lower post-intervention.Step 3: Compared to intervention arm I, the percentage is lower in intervention arms II and III each.

In Step 1, the pre-post-comparisons in intervention arms II and III were tested using a 2.5% significance level (Bonferroni correction). If at least one pre-post-comparison was significant, Step 2 was conducted (significant difference in one arm in step 1: 2.5% significance level, significant difference in both arms: 5% significance level). In case the pre-post-comparison in Step 2 was significant, the pre-post-comparisons between intervention arms were tested in Step 3 to examine intervention effectiveness. Logistic mixed effects models were used to explore the primary objective in the hierarchical procedure. The models considered the nested structure of the data with multiple cases per patient and patients nested in practices by including a random effect in the logistic mixed effects model for patients and practices. As fixed effects, timepoint (pre/post), gender, and age group (<18, 18–65, >65 years) were included in step 1 and 2. In step 3, the intervention arm was added as fixed effect.

The primary outcome was additionally analysed comparing the three intervention arms to matched standard care, using a logistic mixed effects model and taking the clustered structure into account. Standard care was matched on practices level by using a propensity score matching based on a logistic regression model under consideration of the matching variables specialist group, baseline number of cases, postal code (first 3 digits), and region. Considered fixed effects in that model are group (intervention arms vs. matched standard care), time-point (pre/post), gender, age, CCI as indication of health status, nationality, and region, town/countryside. Secondary outcomes were analysed using mixed logistic regression models comparing intervention arms with the matched standard care. All *p*-values for secondary analysis are of explorative nature. Findings reported here focus on the prescribing of antibiotics for upper respiratory tract infections.

## 5. Conclusions

Significantly reduced antibiotic prescribing for non-complicated infections in the intervention arms, compared to matched standard care show the impact of the implementation program. Further research should explore which interventions can increase the effects.

## Figures and Tables

**Figure 1 antibiotics-10-01151-f001:**
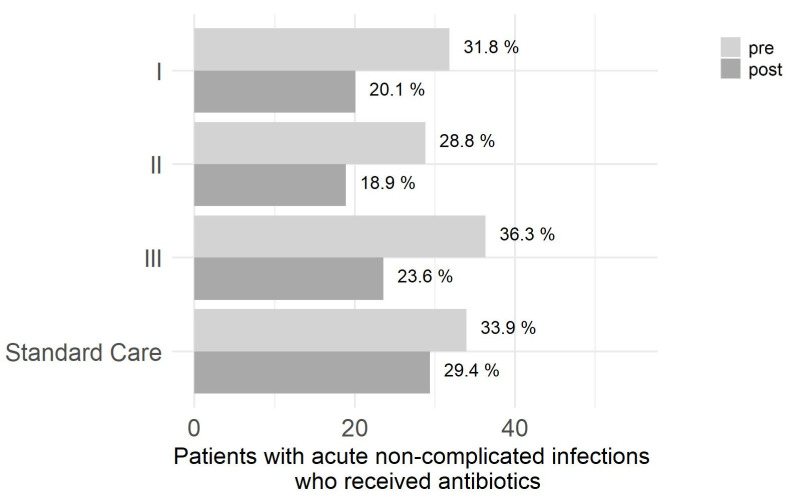
Percentage of patient cases who consulted primary care practices with acute non-complicated infections (index diseases) and were treated with systemic antibiotics (primary outcome). I = intervention arm 1; II = intervention arm II; III = intervention arm III.

**Figure 2 antibiotics-10-01151-f002:**
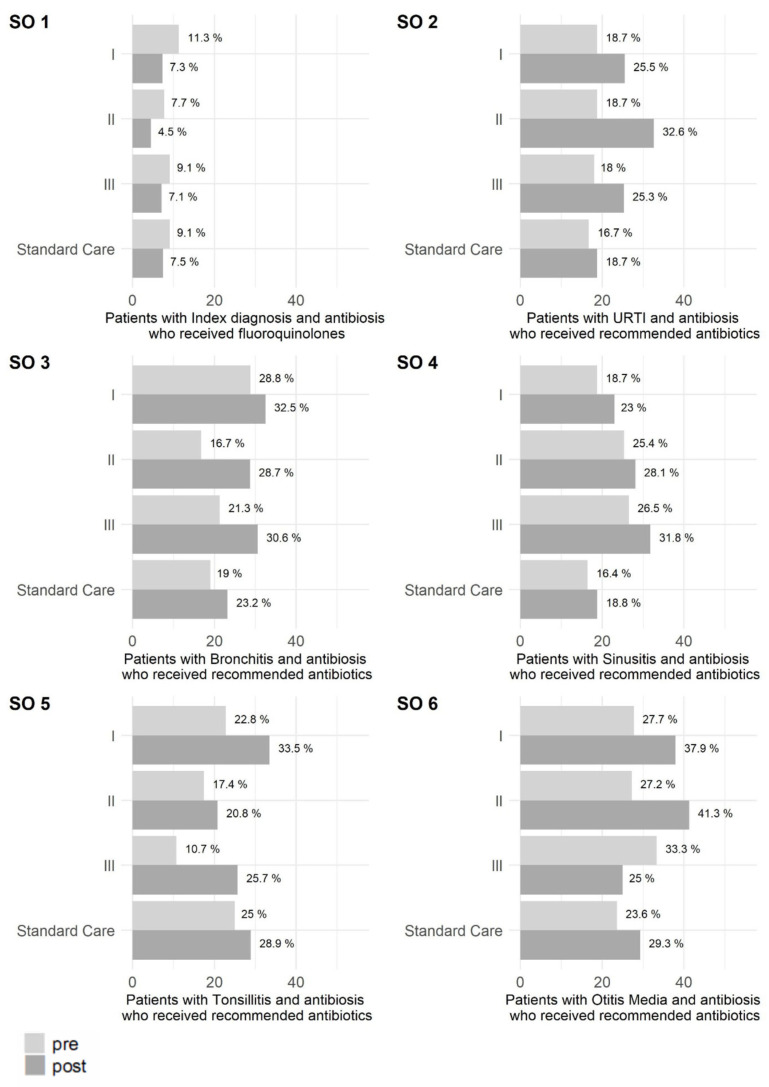
Percentage of patient cases who consulted primary care practices with acute non-complicated infections (index diseases) and received fluoroquinolones or were treated with recommended systemic antibiotics (subgroups). SO = Secondary Outcome.

**Table 1 antibiotics-10-01151-t001:** Patient characteristics of included cases (post-intervention).

		Intervention (Cases)			Matched Standard Care (No Intervention)
		Arm I	Arm II	Arm III	
		(*n* = 10,143)	(*n* = 6730)	(*n* = 5076)	(*n* = 25,144)
Age, *n* (%)	<18	578 (5.7)	757 (11.2)	143 (2.8)	4210 (16.7)
	18–65	7869 (77.6)	4684 (69.6)	3910 (77.0)	18,960 (75.4)
	>65	1696 (16.7)	1289 (19.2)	1023 (20.2)	1974 (7.9)
Gender, *n* (%)	Male	4405 (43.4)	2900 (43.1)	2074 (40.9)	11,891 (47.3)
Nationality, *n* (%)	German	9115 (89.9)	5425 (80.6)	4144 (81.6)	18,681 (74.3)
CCI, *n* (%)	0	5734 (56.5)	3858 (57.3)	2604 (51.3)	17,412 (69.2)
	1, 2	2926 (28.8)	1945 (28.9)	1609 (31.7)	6259 (24.9)
	3, 4	781 (7.7)	453 (6.7)	492 (9.7)	830 (3.3)
	≥5	702 (6.9)	474 (7.0)	371 (7.3)	643 (2.6)
Index diseases, *n* (%)	Bronchitis	2442 (24.1)	1569 (23.3)	1457 (28.7)	5796 (23.1)
	URTI	7620 (75.1)	4916 (73.0)	3719 (73.3)	17,663 (70.2)
	Sinusitis	962 (9.5)	867 (12.9)	700 (13.8)	1642 (6.5)
	Tonsillitis	507 (5.0)	335 (5.0)	220 (4.3)	1654 (6.6)
	Otitis Media	474 (4.7)	357 (5.3)	263 (5.2)	1809 (7.2)

CCI: Charlson Comorbidity Index; URTI: upper respiratory tract infection.

**Table 2 antibiotics-10-01151-t002:** Comparison of antibiotic prescribing rates by mixed logistic regression (pre vs. post, primary outcome) and comparison between intervention arms adjusted for covariates (age, gender).

		OR	95% CI of OR		*p*-Value
			Lower	Upper	
Intervention arm I	post vs. pre	0.523	0.485	0.563	<0.001
	Female vs. male	1.274	1.179	1.376	<0.001
	Age < 18 vs. 18–65	0.675	0.559	0.815	<0.001
	Age > 65 vs. 18–65	1.324	1.193	1.469	<0.001
Intervention arm II	post vs. pre	0.547	0.493	0.607	<0.001
	Female vs. male	1.371	1.227	1.532	<0.001
	Age < 18 vs. 18–65	0.64	0.412	0.994	0.047
	Age > 65 vs. 18–65	1.167	1.008	1.351	0.038
Intervention arm III	post vs. pre	0.519	0.467	0.576	<0.001
	Female vs. male	1.296	1.164	1.443	<0.001
	Age < 18 vs. 18–65	0.676	0.494	0.925	0.014
	Age > 65 vs. 18–65	1.479	1.299	1.684	<0.001
Comparison between intervention arms	Intervention arm II vs. I	0.863	0.658	1.130	0.284
	Intervention arm III vs. I	1.019	0.781	1.331	0.888
	Intervention arm III vs. II	1.182	0.895	1.561	0.239
	post vs. pre	0.561	0.535	0.589	<0.001
	Female vs. male	1.274	1.214	1.338	<0.001
	Age < 18 vs. 18–65	0.683	0.595	0.784	<0.001
	Age > 65 vs. 18–65	1.255	1.179	1.337	<0.001

**Table 3 antibiotics-10-01151-t003:** Comparison of antibiotic prescribing rates in intervention arms to matched standard care by mixed logistic regression (pre vs. post, primary outcome) adjusted for additional covariates.

		OR	95% CI of OR		*p*-Value
			Lower	Upper	
Comparison to standard care	Intervention arm I vs. standard care	0.596	0.572	0.621	<0.001
	Intervention arm II vs. standard care	0.661	0.629	0.695	<0.001
	Intervention arm III vs. standard care	0.726	0.689	0.764	<0.001
	post vs. pre	0.699	0.679	0.721	<0.001
	Female vs. male	1.198	1.162	1.234	<0.001
	Age < 18 vs. 18–65	0.54	0.503	0.579	<0.001
	Age > 65 vs. 18–65	0.836	0.778	0.899	<0.001
	CCI 1 and 2 vs. 0	1.584	1.531	1.638	<0.001
	CCI 3 and 4 vs. 0	1.603	1.498	1.716	<0.001
	CCI ≥ 5 vs. 0	1.503	1.395	1.62	<0.001
	Northern Europe vs. German	0.916	0.740	1.128	0.417
	Eastern Europe, Turkey, Arabic states vs. German	0.978	0.935	1.022	0.321
	Other vs. German	0.818	0.719	0.929	0.002
	Southern Europe vs. German	1.087	0.994	1.189	0.066
	urban vs. rural	0.736	0.712	0.761	<0.001

CCI: Charlson Comorbidity Index; vs.: versus; Northern Europe, Eastern, Europe, Southern Europe, Other: Nationality.

## Data Availability

All analyses generated for this study are included in this manuscript or Appendix A. The original datasets that support the findings of this study are not publicly available due to restrictions stipulated by German law and the data provider. AOK can be contacted for access to the data.

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
