# Peer review of "Assessing Reduction of Antibiotic Prescribing for Acute, Non-Complicated Infections in Primary Care in Germany: Multi-Step Outcome Evaluation in the Cluster-Randomized Trial ARena"

_antibiotics, 2021, doi:10.3390/antibiotics10101151_

Round 1
Reviewer 1 Report
This is an important and interesting manuscript. Therefore, I only have few suggestions for the authors:
line 21 - you have several double spaces throughout the text
line 98 - write also the number of patients, not only cases
line 142 - add the test used under the tables
line 202 - majority of your references are outdated, please compare your results to at least 5-10 references from 2020 od 2021
Author Response
This is an important and interesting manuscript. Therefore, I only have few suggestions for the authors:
*Thank you Reviewer 1 for taking the time to review our manuscript and for your kind assessment. We appreciate the opportunity to improve our manuscript.
line 21 - you have several double spaces throughout the text
All remaining double spaces have been removed.
line 98 - write also the number of patients, not only cases
*We thank you for raising this point. The outcome analysis in ARena is based on quarterly claims data provided by a health insurer. The data reflect patient cases with ARena index diagnoses as recorded by physicians for reimbursement. Since each patient could produce more than one case, cases were focused in the analysis. Our reporting follows the statistical analysis plan that was defined prior to the analysis.
line 142 - add the test used under the tables
*Thank you for pointing this out, the tables and p-values result from the mixed log regression analysis by testing the coefficients to be equal to zero. We added this information to the description of the tables now.
line 202 - majority of your references are outdated, please compare your results to at least 5-10 references from 2020 od 2021
*Thank you for drawing our attention to this. We have added more current references now to the discussion.
Reviewer 2 Report
This is a well-done cluster randomized trial assessing antibiotic prescribing for acute, non-complicated infections in primary care in Germany. I believe this article will be a good add to the current literature on this topic. Please see below for some suggestions for improvement prior to publication.
Line 74 (and others): Quality circle should be better defined here so folks don’t have to look up the published protocol to understand.
Line 78-79: It is a little unclear that the e-learning on communication is for the practice teams as well (and somehow different than the e-learning on communication everyone got). I can see that is the case from looking at the published protocol. And who qualifies as a practice team? Also, QC I assume stands for quality circle, but it is not defined as such.
Line 80-82: This sentence can be improved with some rewording to make it clearer.
Line 117-121: Why are the intervention arms split up this way in these two sentences when there was a significant reduction in all 3?
Line 123-124: All these numbers are in the table so don’t need to duplicate.
Table 2: Perhaps the title should also say adjusted for additional covariates as it was a little confusing looking at it.
Table 3: Are the additional covariates including the overall (pre, post, all interventions, and standard of care)? I assume yes, so if no, it should be better explained.
Line 160-161: You mentioned “lower in all intervention arms;” however, in I vs standard care, the OR is not significant.
Line 166-169: Reword this sentence as it is confusing.
Line 173: Used commas instead of decimals.
Line 190-191: Reword.
Secondary outcomes in general: Are prescribing rates from all office visits or all prescriptions? It is described as percentage of patients with a prescription for recommended antibiotics, but the table/figure makes it look like it might be out of just those patients that got antibiotics? This should be explained in more detail because many of the disease states do not need antibiotics, so a lower rate (out of all office visits) of a recommended antibiotic wouldn’t necessarily be bad.
Figure 2: I assume you meant antibiotics instead of “antibiosis” in all of the descriptions of the figures.
Line 268-274: A little confusing. Perhaps the English translation threw off what you were trying to say.
Line 275-276: Current guidelines recommend antibiotics for uncomplicated viral infections? That doesn’t sound right. You should cite the guidelines. And if they do recommend antibiotics for viral infections, that should be further discussed…
Line 292-293: So you are saying that none of these cases warranted antibiotic therapy? If that is the case, why were recommended antibiotics included?
Line 332: Further explanation on “matched clusters” would be beneficial.
Line 345: That means patients were individually reached out to in order to have their data included?
Study population in general: Explain quarterly matching a little better. Perhaps giving an example would help.
Line 359-368: Recommended antibiotics don’t match the supplementary table A7. Also, percentage of cases should be better explained. Is this percent of cases receiving a fluoroquinolone or recommended antibiotic out of all cases with that diagnosis?
Supplementary Table A3 (and possibly other tables): Please double check the math. I was unable to get the same percentages based on what I assumed the math was supposed to be. For example, for the Primary Outcome in the preintervention for arm 1, the sample size is 9673. From my understanding, the sample size of the fluoroquinolone endpoint should be the same as the number of folks meeting the primary outcome. However, 3119/9673 is 32.2%. Some explanation to why the numbers don’t add up and why some might be excluded, included more than once, etc. might be useful.
Author Response
This is a well-done cluster randomized trial assessing antibiotic prescribing for acute, non-complicated infections in primary care in Germany. I believe this article will be a good add to the current literature on this topic. Please see below for some suggestions for improvement prior to publication.
*Thank you Reviewer 2 for taking the time to assess our manuscript and your valuable comments.
Line 74 (and others): Quality circle should be better defined here so folks don’t have to look up the published protocol to understand.
*We agree with this comment completely and therefore added information here.
Line 78-79: It is a little unclear that the e-learning on communication is for the practice teams as well (and somehow different than the e-learning on communication everyone got). I can see that is the case from looking at the published protocol. And who qualifies as a practice team? Also, QC I assume stands for quality circle, but it is not defined as such.
*Thank you very much for drawing our attention to this. We have added information to the text now to increase transparency.
Line 80-82: This sentence can be improved with some rewording to make it clearer.
*Thank you for this comment. We rephrased to make this sentence clearer.
Line 117-121: Why are the intervention arms split up this way in these two sentences when there was a significant reduction in all 3?
*Thank you for addressing this. We chose to phrase the two sentences to follow the three-step hierarchical test procedure and describe findings accordingly.
Line 123-124: All these numbers are in the table so don’t need to duplicate.
*Thank you for detecting this. You are absolutely right and we deleted this sentence.
Table 2: Perhaps the title should also say adjusted for additional covariates as it was a little confusing looking at it.
*We have added this information now.
Table 3: Are the additional covariates including the overall (pre, post, all interventions, and standard of care)? I assume yes, so if no, it should be better explained.
*Yes, they do.
Line 160-161: You mentioned “lower in all intervention arms;” however, in I vs standard care, the OR is not significant.
*Thank you for this comment. We can follow your train of thoughts and considered adding information here (i.e. p-value), however, we decided against it as this refers to secondary outcomes that were not subject to case number calculation.
Line 166-169: Reword this sentence as it is confusing.
*We re-phrased to avoid confusion, thank you for pointing this out to us.
Line 173: Used commas instead of decimals.
*Thanks, we changed that now.
Line 190-191: Reword.
*Thank you for pointing to this, we changed the wording.
Secondary outcomes in general: Are prescribing rates from all office visits or all prescriptions? It is described as percentage of patients with a prescription for recommended antibiotics, but the table/figure makes it look like it might be out of just those patients that got antibiotics? This should be explained in more detail because many of the disease states do not need antibiotics, so a lower rate (out of all office visits) of a recommended antibiotic wouldn’t necessarily be bad.
*Thank you for this comment. Yes, recommended antibiotics refer to patient cases who got antibiotics, not to all office visits. We added some information to the text to clarify.
Figure 2: I assume you meant antibiotics instead of “antibiosis” in all of the descriptions of the figures.
*Yes, also thank you for this comment. We made the necessary changes.
Line 268-274: A little confusing. Perhaps the English translation threw off what you were trying to say.
*We changed the wording to provide more transparency.
Line 275-276: Current guidelines recommend antibiotics for uncomplicated viral infections? That doesn’t sound right. You should cite the guidelines. And if they do recommend antibiotics for viral infections, that should be further discussed…
*Thank you for raising this point. As we mentioned in the text of our manuscript, current guidelines in Germany recommend to consider antibiotic treatment for non-complicated infections in patients with co-morbidities, leaving the decision to prescribe or not to the discretion of the treating physician. We changed the wording here slightly to increase transparency and added references to the guidelines (as already were provided in the Methods section).
Line 292-293: So you are saying that none of these cases warranted antibiotic therapy? If that is the case, why were recommended antibiotics included?
*Your comment made us aware of a potential lack of transparency, thank you very much for pointing this out. We did not intend to say that, but point to a thorough consideration of ICD-codes that were defined as exclusion criteria. We changed the wording here to provide clarification.
Line 332: Further explanation on “matched clusters” would be beneficial.
*Thank you for finding this phrase which was not meant to be in the text anymore. We deleted it now and changed the wording accordingly.
Line 345: That means patients were individually reached out to in order to have their data included?
*Thank you for addressing this aspect. Patients were not actively recruited, but anonymized cases were automatically included when participating physicians claimed reimbursement for them from the health insurer. All participating physicians consented in the use of their claims data and signed a data release form. Patients in North Rhine-Westphalia had to give an additional written consent. We have added this information to provide more transparency.
Study population in general: Explain quarterly matching a little better. Perhaps giving an example would help.
*ICD-10 coded diagnoses and prescribing were brought together at quarter level since exact dates of prescribing were not available in the provided data. We described this in the Strengths and Limitations section and changed the wording now to clarify.
Line 359-368: Recommended antibiotics don’t match the supplementary table A7. Also, percentage of cases should be better explained. Is this percent of cases receiving a fluoroquinolone or recommended antibiotic out of all cases with that diagnosis?
*Thank you for pointing this out to us, we included the necessary changes. The information that secondary outcomes focus on subgroups of the primary outcome population was already provided in the Results section of the manuscript and we added this information to the description of Figure 2.
Supplementary Table A3 (and possibly other tables): Please double check the math. I was unable to get the same percentages based on what I assumed the math was supposed to be. For example, for the Primary Outcome in the preintervention for arm 1, the sample size is 9673. From my understanding, the sample size of the fluoroquinolone endpoint should be the same as the number of folks meeting the primary outcome. However, 3119/9673 is 32.2%. Some explanation to why the numbers don’t add up and why some might be excluded, included more than once, etc. might be useful.
*Thank you for this comment. We checked the math and are certain that it is correct. Please note that as reported in our manuscript, sample sizes between primary outcome and secondary outcomes were not the same, but differed. Secondary outcomes focused on subgroups of the primary outcome population. The Table provides the respective sample sizes of the endpoint populations and the percentage of cases who received fluoroquinolones or recommended antibiotics. We added a note to the table to clarify.
Reviewer 3 Report
The paper looks good. But the authors need to clarify why the comparison between II vs I, III vs I, and III vs II instead of ANOVA. There is also an error in Figure 2 or in the text line 184. Otitis media are "41.5%" and in figure 2 and the supplement the rate is 41.3%. There is a random "a=1," floating above page 8. One of the limitations that needs to be addressed is the possibility of repeat patients with drug resistant organism requiring repeated antibiotic treatment which may artificially inflate numbers in either groups including the control group.Author Response
Comments and Suggestions for Authors
The paper looks good. But the authors need to clarify why the comparison between II vs I, III vs I, and III vs II instead of ANOVA.
*Thank you for taking the time to review our manuscript and your kind words. The hierarchical three-step design for the analysis provides the explanation for the comparisons. We added information to clarify.
There is also an error in Figure 2 or in the text line 184. Otitis media are "41.5%" and in figure 2 and the supplement the rate is 41.3%.
*Thank you for detecting this typo in the text. We corrected it now.
There is a random "a=1," floating above page 8.
*Thank you also for finding this remnant from the template. We deleted it now.
One of the limitations that needs to be addressed is the possibility of repeat patients with drug resistant organism requiring repeated antibiotic treatment which may artificially inflate numbers in either groups including the control group.
*You are absolutely right in pointing to such a possibility. However, in our analysis, a thorough consideration of coded diagnoses that were to be excluded was undertaken and patient cases such as you described therefore were not included in the analysis.